# Isolation of an Insulin-Like Receptor Involved in the Testicular Development of the Mud Crab *Scylla paramamosain*

**DOI:** 10.3390/ijms241713639

**Published:** 2023-09-04

**Authors:** An Liu, Shuang Hao, Fang Liu, Huiyang Huang, Haihui Ye

**Affiliations:** 1College of Fisheries, Jimei University, Xiamen 361021, China; liuan@jmu.edu.cn (A.L.); liufang@jmu.edu.cn (F.L.); 2College of Ocean and Earth Sciences, Xiamen University, Xiamen 361102, China; 22320191151018@stu.xmu.edu.cn (S.H.); huiyang@xmu.edu.cn (H.H.)

**Keywords:** insulin-like receptor, insulin-like androgenic gland hormone, testicular development, spermatogenesis, mud crab

## Abstract

Insulin-like androgenic gland hormone (IAG) is a key regulator of male sexual differentiation in crustaceans that plays important roles in secondary sexual characteristics and testicular development. As a hormone, IAG interacts with its membrane receptor to initiate downstream signal pathways to exert its biological functions. In this study, we isolated a full-length cDNA of an insulin-like receptor (*Sp-IR*) from the mud crab *Scylla paramamosain*. Sequence analysis revealed that this receptor consists of a Fu domain, two L domains, three FN-III domains, a transmembrane domain, and a tyrosine kinase domain, classifying it as a member of the tyrosine kinase insulin-like receptors family. Our results also suggested that *Sp-IR* was highly expressed in the testis and AG in males. Its expression in the testis peaked in stage I but significantly decreased in stages II and III (*p* < 0.01). Next, both short- and long-term RNA interference (RNAi) experiments were performed on males in stage I to explore *Sp*-IR function in mud crabs. The results showed that *Sp-vasa* and *Sp-Dsx* expression levels in the testis were significantly down-regulated after the specific knockdown of *Sp-IR* by RNAi. Additionally, the long-term knockdown of *Sp-IR* led to a considerable decrease in the volume of seminiferous tubules, accompanied by large vacuoles and a reduced production of secondary spermatocytes and spermatids. In conclusion, our results indicated that *Sp-IR* is involved in testicular development and plays a crucial role in transitioning from primary to secondary spermatocytes. This study provided a molecular basis for the subsequent analysis of the mechanism on male sexual differentiation in Brachyuran crabs.

## 1. Introduction

Sexual determination and differentiation are prominent topics in biological research. Shrimps and crabs, as representative groups of crustacean species, are valuable protein sources for humans and hold significant economic importance in the marine organism industry. It has been observed that there are distinct disparities in growth rate, individual size, and nutritional value between male and female individuals in certain species [1,2]. Therefore, studying the regulatory mechanisms of gender determination and differentiation is a promising approach in the aquaculture industry to enhance production value and develop monosex aquaculture technology specifically for crustacean species.

The androgenic gland (AG) is a unique endocrine gland found in male crustaceans, playing a crucial role in regulating gender differentiation, spermatogenesis, and the maintenance of male secondary sexual characteristics [3,4]. Previous research has shown that AG promotes the rapid formation of secondary spermatocytes from primary spermatocytes in the testis [5]. Subsequently, it was discovered that an insulin-like hormone secreted by the AG is responsible for these functions, leading to its designation as an insulin-like androgenic gland hormone (IAG) [6,7]. Over time, the biological functions of IAG in primary and secondary sexual characteristics have been confirmed in various crustacean species [8]. For example, in the redclaw crayfish (*Cherax quadricarinatus*), silencing the IAG gene resulted in the feminization of some male individuals, evidenced by a reduction in the number of spermatozoa, the degeneration of a significant portion of the testis, and the expression of the vitellogenin gene, which is typically associated with female reproduction [9]. A recent study on the red-striped shrimp, *Lysmata vittata*, which exhibits protandric simultaneous hermaphroditism, demonstrated that inhibiting IAG had significant effects. The knockdown of IAG resulted in the inhibition of both oogenesis and spermatogenesis, as well as the development of abnormal appendices masculinae [10]. Similarly, in the giant freshwater prawn, *Macrobrachium rosenbergii*, the removal of the AG from juvenile males led to sex reversal [3]. Additionally, silencing the *Mr-IAG* gene resulted in AG hypertrophy and the arrest of testicular spermatogenesis [11]. Furthermore, it has been discovered that IAG is crucial for the development of pubertal males into reproductively competent adult males in *M. rosenbergii* [12]. By manipulating the IAG switch during the sexual differentiation process, the successful establishment of all-male or all-female populations in this species has been achieved [13,14].

The insulin-like receptor (IR) is a transmembrane receptor belonging to the tyrosine kinase receptor superfamily. It is initially synthesized as a single chain and undergoes glycosylation and folding to form a mature protein, an α2β2 dimer, which plays a crucial role in transmitting the cascade signals of insulin-like peptides from the extracellular to the intracellular region [15,16]. The first identified putative insulin-like androgenic gland hormone receptor (*Mr-IR*) was isolated from the giant freshwater prawn *M. rosenbergii* [17]. *Mr-IR* is widely expressed in various tissues of both male and female individuals. Knockdown of *Mr-IR* in male individuals through RNA interference (RNAi) did not affect their growth but impeded the development of spermatocytes and led to AG hypertrophy [17]. Further studies revealed that the continuous suppression of *Mr-IR* in males using small interfering RNA (siRNA) induced sex reversal, indicating the involvement of *Mr-IR* in the regulation of sexual differentiation [18]. In the eastern spiny lobster *Sagmariasus verreauxi*, the tyrosine kinase insulin receptor (*Sv-TKIR*) is expressed in the male testis and antennal glands in both sexes [19]. Notably, this receptor can be activated by recombinant IAGs and human insulin in a dose-dependent manner, suggesting that the signaling pathway of this receptor is mediated through the MAPK/ERK pathway. In the Chinese shrimp *Fenneropenaeus chinensis*, *Fc-IAGR* is specifically expressed in males. Prolonged suppression of *Fc-IAGR* gene expression leads to a significant decrease in the amount of sperm cells and results in the arrest of germ cells at the secondary spermatocyte stage [20].

Mud crabs, particularly the dominant species *S. paramamosain* in China, play a significant role in coastal fisheries in tropical and subtropical Asia [21]. In recent years, they have gained importance as cultured species on the southeast coast of China and have become popular in the fishery industries of Southeast Asian countries [22]. In decapod species, having monosex populations in aquaculture offers distinct advantages due to dimorphic growth patterns that result in variations in animal size at harvest [2]. However, the regulatory mechanisms underlying sexual determination and differentiation in mud crabs remain unclear, and the development of sexual control techniques for this species is still a challenging task.

In this study, we first cloned and identified an insulin-like receptor (IR) gene from the mud crab *S. paramamosain*. Based on its expression profile in the testis, we proposed that *Sp*-*IR* may participate in testicular development and spermatogenesis as a putative IAG receptor. Subsequently, both short- and long-term RNAi experiments were performed to validate the function of *Sp*-*IR* in male *S. paramamosain*.

## 2. Results

### 2.1. Cloning and Sequence Analysis of Sp-IR

The full-length cDNA sequence of *Sp-IR* was successfully obtained through a combination of 3′ and 5′ RACE techniques and touchdown PCR. The mRNA sequence of *Sp-IR* in the testis of *S. paramamosain* was found to be 6538 bp in length (GenBank OQ361826). It consisted of a 306 bp 5′ untranslated region (UTR), a 559 bp 3′ UTR, and a substantial open reading frame (ORF) spanning 5673 bp. The ORF encoded a protein consisting of 1890 amino acids (Appendix A).

Bioinformatics analysis revealed that *Sp*-IR belonged to the IR family. The protein could be divided into three distinct regions. The extracellular region comprised a CR domain, two L domains, and three FNIII domains. The transmembrane region was hydrophobic, facilitating its integration into the cell membrane. The intracellular region contained a tyrosine kinase domain, which is crucial for signal transduction and downstream signaling events (Figure 1).

### 2.2. Multiple Sequence Alignment and Phylogenetic Tree Analysis

Figure 2 displays the results of multiple sequence alignment of the tyrosine kinase domain. Among the species compared, *Sp*-*IR* exhibits the highest similarity with the conserved tyrosine kinase domain of *F. chinensis*, with a similarity of 81.10%. It also shows a relatively high similarity of 74.32% with the tyrosine kinase domain of *S. verreauxi*. Comparing *Sp*-*IR* with other species, including *M. rosenbergii*, *Bombyx mori*, *Drosophila melanogaster*, *Aedes aegypti*, and other arthropods, the amino acid sequence similarity exceeds 40% in all cases.

The phylogenetic tree analysis results are presented in Figure 3. Initially, *Sp*-*IR* clustered with *Fc*-IAGR in *F. chinensis*, *Pv*-IR in *Penaeus vannamei*, and *Sv*-IR in *S. verreauxi*. Subsequently, it formed a cluster with *Mr*-IR in *M. rosenbergii*, indicating a high degree of similarity between these IRs.

### 2.3. Tissue Distribution of Sp-IR in Male S. paramamosain

The expression pattern of *Sp*-*IR* in various tissues of male *S. paramamosain* at stage II was assessed using RT-PCR. The results revealed that *Sp-IR* exhibited high expression levels in the testis and AG (Figure 4). However, no transcript of *Sp*-*IR* was detected in the other tissues examined.

### 2.4. Profile Expression of Sp-IR in Testicular Development

To investigate the potential involvement of *Sp*-*IR* in the reproductive process of male *S. paramamosain*, the expression profile of *Sp*-*IR* was analyzed across three testis’ developmental stages using qRT-PCR. The results demonstrated that *Sp-IR* exhibited the highest expression level at stage I (Figure 5). However, a significant decrease in *Sp*-*IR* expression was observed in both stage II and stage III (*p* < 0.05).

### 2.5. Expression of Sp-IR and Related Genes after Short-Term Injection of Sp-IR dsRNA

In studying sex determination and differentiation mechanisms in crustaceans, researchers often rely on homologous marker genes identified in model organisms, such as *vasa, Dsx*, *Sxl*, *Tra*, *Dmrt*, and *Sox-5* genes [23]. Vasa is a well-known marker gene for germ cells due to its specific expression and essential functions in gametogenesis [24]. It has been confirmed in many organisms that vasa play crucial roles in spermatogenesis [25,26,27]. In male *S. paramamosain*, *Sp-vasa* was specifically expressed in the testis, and its mRNA was detected exclusively in germ cells [28]. The expression of *Sp-vasa* gradually increased from spermatogonia to spermatocytes but was absent in spermatozoa, indicating its involvement in spermatogenesis before the formation of spermatozoa. Similarly, the expression pattern of the Dsx gene in the testis of *F. chinensis* is comparable to that of *Sp-vasa* [29]. Silencing of *Fc-Dsx* was shown to inhibit *Fc-IAG* gene expression, suggesting the involvement of Dsx in IAG-regulated spermatogenesis [29]. Therefore, in this study, we utilized *Sp-vasa* and *Sp-Dsx* as indicators of spermatogenesis.

To assess the effectiveness of *Sp-IR* knockdown in the testis, male crabs at stage I were subjected to RNAi by injecting *Sp*-*IR* dsRNA. The results demonstrated a significant reduction in the expression level of *Sp-IR* after *Sp*-*IR* dsRNA injection (*p* < 0.01). The knockdown efficiency reached up to 85.97% (Figure 6A). Furthermore, the silencing of *Sp-IR* resulted in a substantial decrease in the expression levels of *Sp-vasa* (Figure 6B) and *Sp-Dsx* (Figure 6C) genes in the testis (*p* < 0.01). In contrast, the injection of *GFP* dsRNA had no significant effect on the expression levels of *Sp*-*IR*, *Sp-vasa* or *Sp-Dsx*.

### 2.6. Effects on Related Genes and Testicular Development after Long-Term Injection of IR dsRNA

To further investigate the role of *Sp*-*IR* in spermatogenesis and testicular development in *S. paramamosain*, a 22-day RNAi experiment was conducted on male individuals at stage I. At the end of the experiment, the interference efficiency and the expression levels of testis development-related genes were assessed. The results demonstrated a significant down-regulation of *Sp*-*IR* expression after 22 days of *Sp-IR* dsRNA injection compared to the crab saline control group (*p* < 0.001). The gene knockdown efficiency was calculated to be 75.83% (Figure 7A). Similarly, the expression levels of *Sp-vasa* (Figure 7B) and *Sp-Dsx* (Figure 7C) transcripts were significantly reduced following the injection of *Sp-IR* dsRNA (*p* < 0.01). In contrast, the injection of *GFP* dsRNA had no notable effect on the expression levels of *Sp-IR*, *Sp-vasa*, or *Sp-Dsx* transcripts in the testis.

Moreover, the histological analysis revealed that the knockdown of *Sp-IR* had a detrimental effect on spermatogenesis and testis development in *S. paramamosain*. In the pre-injection control group, the lumen of the seminiferous tubules contained a substantial number of primary spermatocytes and spermatids, indicating active spermatogenesis (Figure 8A). The crab saline control group and the *GFP* dsRNA group showed similar proportions of the four types of cells compared to the pre-injection control group, with only a few spermatids transitioning into spermatozoa (Figure 8B,C). In contrast, the injection of *Sp-IR* dsRNA resulted in irregularly shaped seminiferous tubules and a significant reduction in their volume. Large vacuoles were also observed in the testis. Furthermore, most of the seminiferous tubules were occupied by spermatogonia and primary spermatocytes, with only a small number of secondary spermatocytes present. No spermatids or mature sperm were observed (Figure 8D).

## 3. Discussion

The understanding of male sexual differentiation in crustaceans has been predominantly attributed to the role of IAG derived from the AG [30,31]. However, the precise mechanism underlying this regulation remains to be fully elucidated. Recent advances have shed light on discovering IAG receptors, offering new insights into the sexual differentiation and testicular development of crustacean species [17,18,19]. In this study, we successfully isolated a cDNA encoding the IR (*Sp*-*IR*) from the mud crab *S. paramamosain*, and through RNAi technology, we investigated the specific functions of *Sp*-*IR* in spermatogenesis and testicular development.

The *Sp*-*IR* gene identified in the mud crab *S. paramamosain* was found to be 1890 aa in length. It consists of various structural domains, including a FU domain, two L domains, three FN-III domains, a transmembrane domain, and a tyrosine kinase domain, consistent with other members of the tyrosine kinase receptor superfamily [32]. Multiple sequence alignment analysis revealed a high degree of homology in the amino acid sequence of *Sp*-*IR* with IR genes reported in other species. Specifically, the sequence similarity of *Sp*-*IR* with the insulin-like receptor genes of *S. verreauxi* and *F. chinensis* in the tyrosine kinase domain exceeded 70% [19,20]. Furthermore, phylogenetic tree analysis demonstrated a close genetic relationship between *Sp*-*IR* and other decapod crustaceans. These findings suggest that IRs in crustacean species are highly conserved, and the identified *Sp*-*IR* gene may serve as a putative IAG receptor in the mud crab. However, further functional assays are necessary to confirm whether this putative receptor can indeed be activated by IAG.

RT-PCR results revealed that *Sp-IR* exhibited high expression levels in the testis and AG of male *S. paramamosain*, consistent with the findings reported in *F. chinensis* and *S. verreauxi*, where *Fc-IAGR* and *Sv-IR* were also highly expressed in the testis [19,20]. Interestingly, in *M. rosenbergii*, the IR gene showed high expression in the female ovary and antennal gland, while its expression was low in the male AG, cerebral ganglion, and thoracic ganglion of both male and female individuals [17]. The high expression of *Sp-IR* in the testis of *S. paramamosain* indicated its potential involvement in testicular development. The expression profile of *Sp*-*IR* mRNA during testicular development was investigated by qRT-PCR. The findings revealed that the expression of *Sp-IR* transcript was highly abundant at stage I but significantly decreased at stages II and III (Figure 5), opposite to that observed for *Fc-IAGR* in *F. chinensis* and *Sp-IAG* in *S. paramamosain*, which showed higher expression at later stages of testicular development [20,33]. It reported that *Fc-IAGR* mainly expressed in secondary spermatocytes and spermatids in *F. chinensis* [20], and *Mr-IR* was located only in spermatocytes of *M. rosenbergii* [17], indicating a possible function of *Fc*-IAGR and *Mr*-IR in the late stage of spermatogenesis. In *S. paramamosain*, it suggests that *Sp-IR* may play a role in the early stages of testicular development, possibly in the differentiation of primary or secondary spermatocytes.

To further confirm the roles of *Sp*-*IR* in the testicular development of male mud crabs, we conducted both short- and long-term RNAi experiments. The results demonstrated that RNAi effectively down-regulated *Sp-IR* expression in the testis. Following the knockdown of *Sp-IR*, there was a significant decrease in the levels of *Sp-vasa* and *Sp-Dsx* transcripts, indicating the disruption of spermatogenesis. Moreover, substantial inhibition of testicular development was observed, characterized by a significant shrinkage in the volume of the seminiferous tubules and the presence of large vacuoles within the seminiferous lobules. These findings strongly suggest that *Sp*-*IR* plays a crucial role in the development of the testis in male mud crabs. The down-regulation of *Sp-vasa* is likely to be responsible for the observed phenotype, as vasa are known to be involved in the differentiation of germ cells into gametes [34]. Indeed, loss of vasa function in the mouse affects differentiation of the male germ cells, resulting in male sterility [35]. Studies in vertebrates have shown that insulin-like growth factor plays an important role in gonadal development, sex determination, and gender differentiation [36]. Knockdown of insulin-like growth factor and its receptor in mice leads to reduced testis size, spermatogenic cell numbers, and sperm production [37]. Similar results have been reported in crustaceans such as the giant freshwater prawn *M. rosenbergii* and the Chinese shrimp *F. chinensis* [17,20]. In *F. chinensis*, silencing the *FcIAGR* gene results in delayed spermatogenesis and the arrest of germ cells at the secondary spermatocyte stage [20]. In *M. rosenbergii*, silencing the *Mr-IR* gene leads to a decrease in sperm cell numbers and a delay in spermatogenesis at the secondary spermatocyte stage [17]. In contrast, in *S. paramamosain*, most sperm cells are arrested at the primary spermatocyte stage after *Sp-IR* knockdown, suggesting that *Sp-IR* may be involved in the transition of primary spermatocytes to secondary spermatocytes. The high expression of *Sp-IR* in spermatogonia, as revealed by in situ hybridization, may explain the observed differences between *S. paramamosain* and *F. chinensis*. In this regard, we have also identified another IR that is highly expressed in stage III testis (unpublished results), suggesting that the regulation of spermatogenesis by insulin-like signaling in *S. paramamosain* is likely to be more complex.

In conclusion, this study identified and characterized an insulin-like receptor (*Sp*-*IR*) in the mud crab *S. paramamosain*. *Sp*-*IR* was found to be highly expressed in the testis and AG and was involved in the transition of primary spermatocytes to secondary spermatocytes and testicular development. As a putative receptor for IAG, the findings provide valuable insights into the mechanism underlying male sexual differentiation in Brachyuran crabs. Further research is needed to elucidate the precise role of *Sp*-*IR* and its interaction with IAG in regulating sexual differentiation.

## 4. Materials and Methods

### 4.1. Animals

Mud crabs (*Scylla paramamosain*) were purchased from a local fisheries wholesale market in the Si-Ming District of Xiamen (Fujian, China). The male individuals selected for the experiments exhibited healthy appendages and good vitality and were at the intermolt stage. Prior to the start of the experiments, the crabs were acclimated in tanks containing filtered seawater at a temperature of 27 ± 2 °C and salinity of 27 ± 0.5 ppt for 5 days and fed with fresh Manila clams (*Ruditapes philippinarum*) daily. Any uneaten bait was promptly removed, and the tanks were replenished with fresh seawater containing sufficient oxygen the following morning.

The development of *S. paramamosain* testis can be divided into five stages, which closely align with the staging system of the Chinese mitten crab *Eriocheir sinensis* [38,39]. In stage I, the testis is transparent, and the seminiferous tubules are relatively small, measuring approximately 105 μm in diameter. This stage is characterized by a high proportion of spermatogonia and primary spermatocytes, with occasional presence of secondary spermatocytes and spermatids. In stage II, the testis appears semitransparent with white coloration. The seminiferous tubules are larger than in stage I, with a diameter of around 210 μm. Most of the germ cells at this stage are spermatocytes. In stage III, the testis takes on a white or milk-white coloration, and the seminiferous tubules further increase in size, measuring approximately 4–6 mm in diameter. Spermatids become the prominent germ cells at this stage. In stage IV, mature sperms become predominant in the testis, and the lumen of the seminiferous tubules continues to expand. In stage V, the testis exhibits a faint yellow color, the seminiferous tubules are devoid of sperm, and the germinal zone undergoes degradation, shrinkage, or even disappear.

Since the differentiation of spermatogonia and spermatocytes occurs before stage IV, the crabs in stages I to III were selected to analyze *Sp-IR* expression. The crabs used for the analysis had the following characteristics: stage I (body weight: 35.67 ± 4.82 g, carapace width: 4.20 ± 0.51 cm), stage II (body weight: 108.24 ± 15.68 g, carapace width: 6.50 ± 0.58 cm), and stage III (body weight: 274.52 ± 18.73 g, carapace width: 8.70 ± 0.45 cm). The crabs were anesthetized by placing them on ice, and testis samples were collected from males at each stage (*n* = 4–5) for *Sp-IR* expression profile analysis using real-time quantitative PCR (qRT-PCR). The samples were then either immediately used for RNA extraction or stored in liquid nitrogen.

### 4.2. Total RNA Extraction and Molecular Cloning

Total RNA was extracted from the samples using TRIzol (Invitrogen, Carlsbad, CA, USA) following the manufacturer’s instructions. The concentration and quality of the extracted RNA were determined using a Q6000 spectrophotometer (Thermo Scientific, Waltham, MA, USA). The integrity of the RNA was evaluated by performing 1.5% (*w/v*) agarose gel electrophoresis. First-strand cDNA synthesis was conducted for subsequent experiments using RevertAid™ First Strand cDNA Synthesis Kit (Thermo Scientific).

A partial cDNA sequence encoding an insulin-like receptor was initially identified from the transcriptome database of *S. paramamosain*. Subsequently, the full-length sequence was amplified using specific primers IR-F and IR-R. To obtain the complete sequence of *Sp-IR*, 3′-RACE and 5′-RACE experiments were performed using the SMARTer™ RACE cDNA Amplification Kit (Clontech, Mountain View, CA, USA) following the manufacturer’s protocol. The PCR products obtained were separated by 1.5% (*w/v*) agarose gels, purified using an agarose gel purification and extraction kit (Aidlab, Beijing, China), and ligated into a pMD19-T vector (Takara, Ohtsu, Japan) for further sequencing. Then, the full-length cDNA sequence of *Sp-IR* was assembled from the obtained sequencing results. The primers utilized in this study are listed in Appendix A.

The open reading frame (ORF) and amino acid sequence of *Sp*-IR were predicted using the ORF finder tool. The conserved domains of *Sp*-IR were predicted using the SMART website (http://smart.embl-heidelberg.de/; accessed on 7 November 2022), while the transmembrane domain of *Sp*-IR was predicted using an online website (http://www.cbs.dtu.dk/services/TMHMM/, accessed on 7 November 2022). To determine the presence of a protein signal peptide, the amino acid sequence was submitted to the SignalP 5.0 Server (http://www.cbs.dtu.dk/services/SignalP/; accessed on 7 November 2022). To assess the homology of the amino acid sequence of *Sp*-IR with those of other species, BlastX homology searches were performed using the NCBI database (http://blast.ncbi.nlm.nih.gov/Blast.cgi; accessed on 8 November 2022). Multiple sequence alignment of the tyrosine kinase domain was conducted using Cluster X2.0 and GeneDoc2.7.000 software. Phylogenetic trees were constructed using the neighbor-joining (NJ) method implemented in MEGA 7.0.26 software, with bootstrap sampling repeated 1000 times to assess the reliability of the tree.

### 4.3. Tissue Distribution of Sp-IR in the Male Mud Crab

To determine the tissue distribution of *Sp-IR* in male mud crabs at stage II, reverse transcription PCR (RT-PCR) was performed. The cDNA templates were generated from 11 different tissues, including the eyestalk ganglion, cerebral ganglion, thoracic ganglion, heart, muscles, hepatopancreas, stomach, epidermis, testis, androgenic gland, and Y-organ. PCR was performed using the following conditions: an initial denaturation step at 95 °C for 3 min, followed by 36 cycles of denaturation at 95 °C for 30 s, annealing at 58 °C for 30 s, and extension at 72 °C for 30 s. The final extension step was performed at 72 °C for 10 min. Negative controls, using water as a template, were included to ensure the absence of contamination. The amplification of *β-actin* was used as the internal control for normalization purposes. After amplification, the PCR products were visualized by electrophoresis on a 1.5% agarose gel and imaged using a UV gel imager (Thermo Scientific, USA). The experiment was repeated thrice.

### 4.4. Short-Term RNA Interference Experiment In Vivo

#### 4.4.1. dsRNA Preparation

To synthesize dsRNA, T7 and SP6 RNA polymerase from Takara were used under the following conditions: incubation at 37 °C for 2 h, followed by a heat inactivation step at 70 °C for 10 min, and a final incubation at 37 °C for 20 min. The synthesized dsRNA, including *IR* dsRNA and *GFP* dsRNA, was analyzed by electrophoresis on a 1.5% agarose gel to confirm the presence and integrity of the dsRNA. The concentrations of the dsRNA samples were then measured using a NanoDrop 2000 spectrophotometer from Thermo Fisher Scientific and stored at −80 °C until further use.

#### 4.4.2. RNA Interference

For this experiment, male mud crabs at stage I with a body mass of 36.72 ± 2.84 were used. Prior to injection, the crabs were temporarily housed in the laboratory for approximately 5 days. They were then randomly divided into three groups (*n* = 10). The first group of crabs was injected with *IR* dsRNA (1 μg/g body mass) prepared in 100 μL crab saline. The second group received only 100 μL of crab saline, while the third group was injected with *GFP* dsRNA (1 μg/g body mass) prepared in 100 μL crab saline. Injections were performed once daily using a microsyringe (Hamilton, Bonaduz, Switzerland) with a thin needle inserted into the arthrodial membrane at the fifth swimming leg. Approximately 48 h after the second injection, the crabs were anesthetized by placing them on ice. Testis samples were dissected and collected for gene expression analysis. Specifically, the expression levels of *Sp-IR*, *Sp-vasa*, and *Sp-Dsx* genes were assessed using quantitative real-time PCR (qRT-PCR).

### 4.5. Long-Term RNA Interference Experiment In Vivo

To investigate the impact of *Sp-IR* on testis differentiation and development in the mud crab *S. paramamosain*, a long-term RNAi experiment was conducted on male crabs at stage I. The crabs used in the experiment had a body mass of 37.56 ± 3.68 g and were temporarily housed in the laboratory for 5 days. Before the injection, six individuals were randomly selected as the pre-injection control group (*n* = 6). The remaining crabs were divided into three groups (*n* = 30). The control group received an injection of 100 μL of crab saline. The other two groups were injected with *Sp-IR* dsRNA and *GFP* dsRNA (1 μg/g body mass) prepared in 100 μL of crab saline, respectively. The injections were performed once every five days, and on day 22 of the experiment, approximately 48 h after the fourth injection, the crabs were anesthetized by placing them on ice and then euthanized. Testis samples were dissected from the crabs for gene expression analysis of *Sp*-*IR*, *Sp-vasa*, and *Sp-Dsx,* using techniques such as qRT-PCR. Additionally, histological analysis of the testis was conducted through hematoxylin and eosin staining.

### 4.6. Real-Time Quantitative PCR (qRT-PCR)

The relative expression levels of the target genes were determined using qRT-PCR. The reactions were carried out on a 7500 Fast Real-Time PCR Detection System (Applied Biosystems, Carlsbad, CA, USA) with a total reaction volume of 20 μL. Each reaction included 2 μL of diluted cDNA, 10 μL of 2× PCR Master Mix with SYBR GREEN (Thermo Scientific, Vilnius, Lithuania), 0.5 μL each of forward and reverse primers (1 mM), and 7 μL of water. The thermal profile for the qRT-PCR reaction consisted of an initial incubation at 95 °C for 3 min, followed by 40 cycles of amplification with the following conditions: denaturation at 95 °C for 10 s, annealing at 58 °C for 30 s, and extension at 72 °C for 30 s. Each sample was analyzed in triplicate, and the *β-actin* gene was simultaneously amplified as an internal control to normalize the data. The primers used for the qRT-PCR are listed in Appendix A. The relative transcript abundance of the target genes was calculated using the 2^−ΔΔCt^ method.

### 4.7. Statistical Analysis

The data are presented as mean ± SEM (standard error of the mean). Statistical analysis was performed using the SPSS v20.0 software. Levene’s test was performed to examine the homogeneity of data variance. If the variance was found to be homogeneous, significance analysis was performed using one-way analysis of variance (ANOVA), followed by Duncan’s test or Dunnett multiple comparison test. Statistical significance was set at *p* < 0.05.

## Figures and Tables

**Figure 1 ijms-24-13639-f001:**
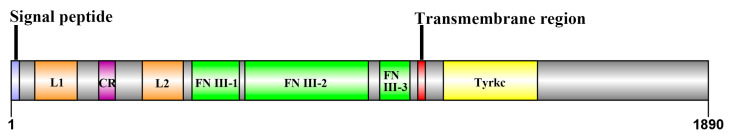
Schematic representation of *Sp*-IR. This receptor was 1890 aa long and comprised two L domains, three FN-III domains, a transmembrane region and a tyrosine kinase domain. L1 and L2: ligand binding domains; CR: cysteine-rich domain; FN III: Fibronectin type III domain; TyrKc: Tyrosine kinase domain.

**Figure 2 ijms-24-13639-f002:**
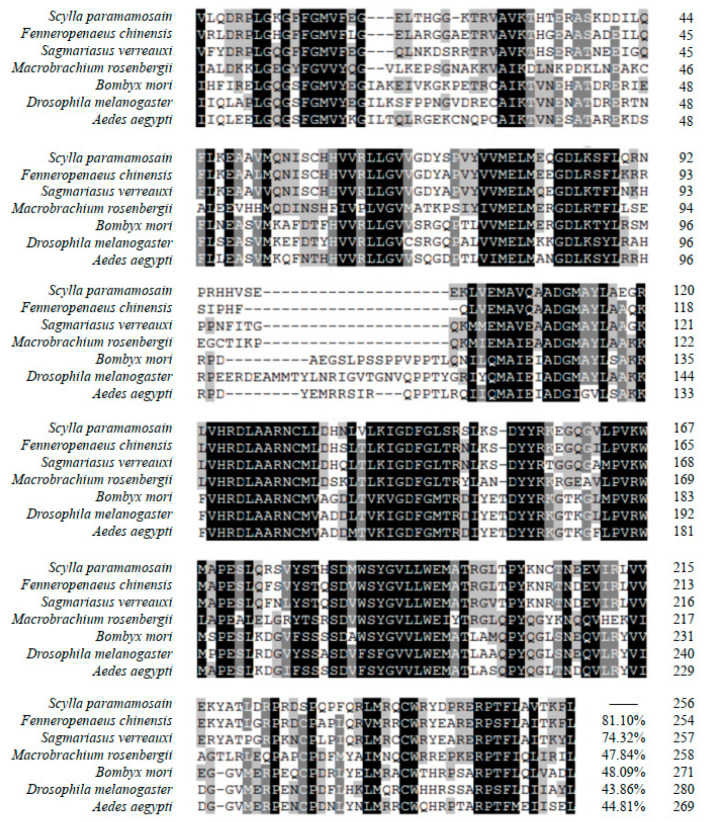
Multiple sequence alignments of the tyrosine kinase domain of *Sp-IR* with that of the other insulin-like receptors. The species and GenBank accessions of amino acid sequences for alignment are as follows: *Fenneropenaeus chinensis* (AVU05021.1); *Sagmariasus verreauxi* (ANC28181.1); *Macrobrachium rosenbergii* (AKF17681.1); *Bombyx mori* (AAF21243.1); *Drosophila melanogaster* (AAC47458); *Aedes aegypti* (AAB17094). The black regions represent highly conserved protein sites.

**Figure 3 ijms-24-13639-f003:**
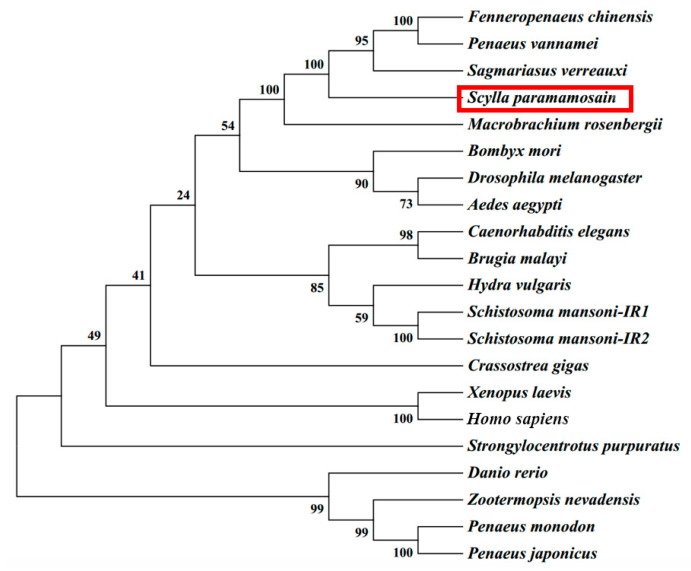
Phylogenetic analysis of *Sp-IR* and homologous genes from other species using neighbor-joining. The *Sp-IR* in this study is highlighted by a box in red, and the node value represents its corresponding neighbor-joining (NJ) value. The species and GenBank accessions for phylogenetic tree analysis are as follows: *Fenneropenaeus chinensis* (AVU05021.1); *Penaeus vannamei* (XP_027207730.1); *Sagmariasus verreauxi* (ANC28181.1); *Macrobrachium rosenbergii* (AKF17681.1); *Bombyx mori* (AAF21243.1); *Drosophila melanogaster* (AAC47458); *Aedes aegypti* (AAB17094); *Caenorhabditis elegans* (AAC47715); *Brugia malayi* (AAW50597); *Hydra vulgaris* (Q25197.1); *Schistosoma mansoni* 1 (AAN39120.1); *Schistosoma mansoni* 2 (AAV65745.2); *Crassostrea gigas* (CAD59674); *Xenopus laevis* (AAC12942.1); *Homo sapiens* (AAI17173.1); *Strongylocentrotus purpuratus* (ABC61312); *Danio rerio* (AAL05594.1); *Zootermopsis nevadensis* (XP_021930936.1); *Penaeus monodon* (XP_037777825.1); *Penaeus japonicus* (XP_042864161.1).

**Figure 4 ijms-24-13639-f004:**
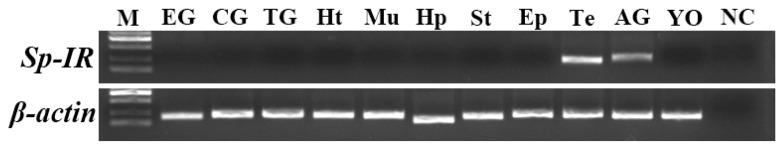
Tissue distribution of *Sp-IR* in male *S. paramamosain*. M: DNA marker; EG: eyestalk ganglion; CG: cerebral ganglion; TG: thoracic ganglion; Ht: heart; Mu: muscle; Hp: hepatopancreas; St: stomach; Ep: epidermis; Te: testis; AG: androgenic gland; YO: Y-organ; NC: amplification of water was set as the negative control.

**Figure 5 ijms-24-13639-f005:**
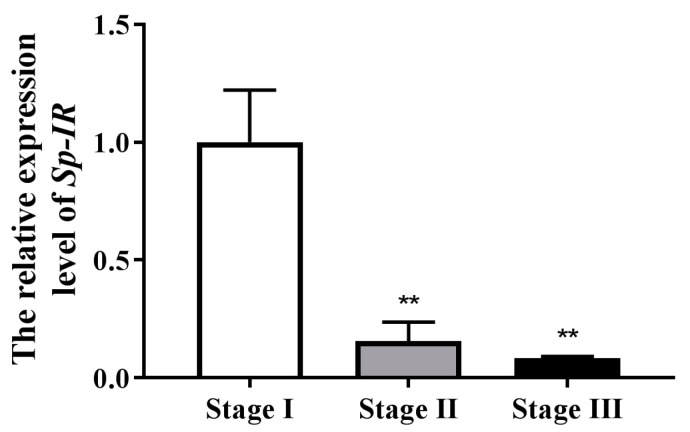
Expression profile of *Sp-IR* in the testicular development of *S. paramamosain*. The data are shown as mean ± SEM (*n* = 4~5); “**” (*p* < 0.01) indicates extremely significant differences from stage I; *β-actin* was used as the reference control gene.

**Figure 6 ijms-24-13639-f006:**
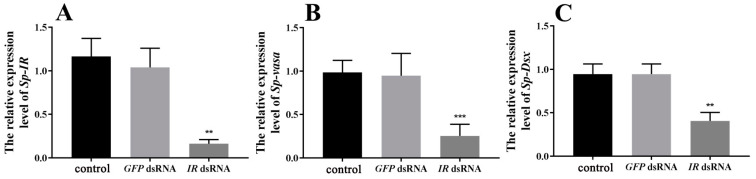
Effects of short-term *Sp-IR* silencing on the expression of *Sp-IR* (**A**), *Sp-vasa* (**B**) and *Sp-Dsx* (**C**) in the testis of *S. paramamosain.* The data are shown as mean ± SEM (*n* = 5~7). “**” (*p* < 0.01) and “***” (*p* < 0.001) indicate significant differences from the crab saline control group.

**Figure 7 ijms-24-13639-f007:**
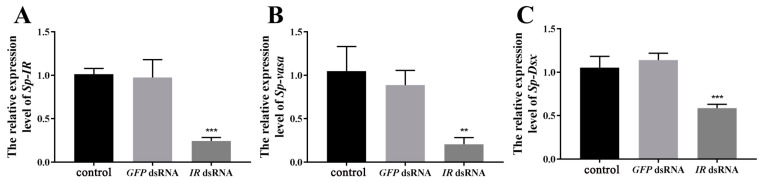
Effects of long-term *Sp-IR* silencing the expression of *Sp-IR* (**A**), *Sp-vasa* (**B**) and *Sp-Dsx* (**C**) in the testis of *S. paramamosain*. The data are shown as mean ± SEM (*n* = 8~12); “**” (*p* < 0.01) and “***” (*p* < 0.001) indicate extremely significant differences from the crab saline control group; *β-actin* was used as a reference control gene.

**Figure 8 ijms-24-13639-f008:**
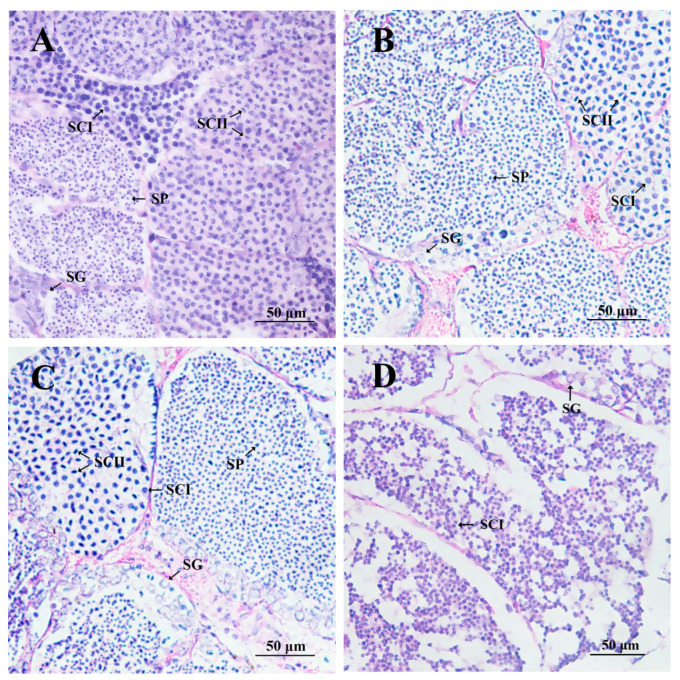
Histological changes in the testis of *S. paramamosain* at stage I in response to long-term injection of *Sp*-*IR* dsRNA. (**A**) pre-injection control; (**B**) Crab saline control; (**C**) *GFP* dsRNA group; (**D**): *Sp*-*IR* dsRNA group. SG: spermatogonia; SCI: primary spermatocytes; SCII: secondary spermatocytes; SP: spermatids.

## Data Availability

The data presented in this study are openly available in GenBank [OQ361826].

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
