# Peer review of "Isolation of an Insulin-Like Receptor Involved in the Testicular Development of the Mud Crab Scylla paramamosain"

_ijms, 2023, doi:10.3390/ijms241713639_

Round 1

Reviewer 1 Report

This was a very interesting paper to review. As discussed by the authors, the understanding of male sexual differentiation in crustaceans has been predominantly attributed to the role of an insulin-like androgenic gland hormone (IAG) derived from the androgenic gland (AG). However, the precise mechanism underlying this regulation remains to be fully elucidated. Recent advances (2016-2020) have shed light on this problem by discovering the insulin-like IAG receptors, members of the receptor tyrosine kinase family, offering new insights into the sexual differentiation and testicular development of crustacean species. In this study, the authors successfully isolated a cDNA encoding the insulin-like receptor from the mud crab S. paramamosain (Sp-IR), and through RNAi technology, investigated the speciific functions of Sp-IR in spermatogenesis and testicular development.

This is an exciting contribution to the recent development of knowledge regarding the structure and function of the IAG, known for over 60 years but only recently identified as part of the evolutionarily conserved insulin signaling system. I believe this crustacean IAG field will provide fundamental new information on the structural biology and evolution of this signaling system since despite general homology, there are radical differences between the secretory origin of the peptide ligand and the end points of receptor activation as compared to e.g. vertebrates and insects.

The study is rationally designed and the paper is well written and convincing. I have only minor comments and suggestions.

Page 1, title, authors list: replace “and” with a comma between 1 and *.

Page 2, line 68: reference 15. The paper by Axel Ullrich’s group should also be cited: Ullrich, A et al., Nature 1985, 313, 756-761.

Page 2, line 77: “Notably this receptor [from the Eastern spiny lobster] can be activated by recombinant IAGs and human insulin…”: the fact that human insulin was only 9 times less potent than the lobster IAG but in fact more potent than some other crustacean IAGs was not discussed or explained in the Aizen paper (reference 18). It could be of interest to show a figure (maybe in supplementary materials) showing a sequence comparison of human insulin and the crustacean IAGs, and evaluate the conservation of the well-known human insulin receptor binding residues in the IAGs (this was not done in the Aizen paper). I can understand if the authors think that this is outside of the scope of their paper since they don’t show actual activation of their receptor by IAG and state in the discussion that further functional assays are necessary.

Page 3, line 107 and Figure 1: what the authors call a FU domain is not incorrect but has been generally defined in the insulin receptor literature as CR domain. It should be spelled that FU means “furin like-repeats, cysteine-rich domain”. The FU domain is not mentioned in the legend of Figure 1.

Page 4, figure 2: it would have been nice to include the human insulin receptor in the sequence comparison.

Page 5, Figure 3: Homo sapines should be Homo sapiens.

Page 5, figure 4: Define column M in the legend.

Page 6, section 2.5. It is not explained what Sp-vasa and Sp-Dsx are.  I would suggest to move lines 251-263 from the discussion to the beginning of section 2.5 so that the purpose and meaning of the experiment is clearer.

Page 9, line 289: “…we have also identified another insulin-like receptor that is highly expressed in stage 3 testis,…”: this should be referenced if published or in press, or otherwise mentioned as submitted or unpublished results.

Author Response

Page 1, title, authors list: replace “and” with a comma between 1 and *.

Response: change has been made accordingly to your suggestion. Please see line 4. Thank you.

Page 2, line 68: reference 15. The paper by Axel Ullrich’s group should also be cited: Ullrich, A et al., Nature 1985, 313, 756-761.

Response: we have supplemented this paper in the reference. Please see line 68 and 481.

Page 2, line 77: “Notably this receptor [from the Eastern spiny lobster] can be activated by recombinant IAGs and human insulin…”: the fact that human insulin was only 9 times less potent than the lobster IAG but in fact more potent than some other crustacean IAGs was not discussed or explained in the Aizen paper (reference 18). It could be of interest to show a figure (maybe in supplementary materials) showing a sequence comparison of human insulin and the crustacean IAGs, and evaluate the conservation of the well-known human insulin receptor binding residues in the IAGs (this was not done in the Aizen paper). I can understand if the authors think that this is outside of the scope of their paper since they don’t show actual activation of their receptor by IAG and state in the discussion that further functional assays are necessary.

Response: It is a nice suggestion. As your mentioned that it is outside of the scope of this paper, so we did not supplement this result. However, we will to make a comparison of human insulin and the crustacean IAGs, and the receptors in our near future study. It is an interesting thing to elucidate the function of insulin and IAG in the aspect of structural biology and evolution. Your comment provides new idea for us. Thank you very much!     

Page 3, line 107 and Figure 1: what the authors call a FU domain is not incorrect but has been generally defined in the insulin receptor literature as CR domain. It should be spelled that FU means “furin like-repeats, cysteine-rich domain”. The FU domain is not mentioned in the legend of Figure 1.

Response: yes, you are right. We have revised to CR domain, and the legend of Figure 1 has also been revised accordingly. Thank you! Please see section 2.1 and line 106 and 114.    

Page 4, figure 2: it would have been nice to include the human insulin receptor in the sequence comparison.

Response: Thanks for your suggestion. In our opinion, it is not necessary to included the human insulin receptor sequence here. The purpose of figure 2 is to show the identity of Sp-IR to arthropod IRs, in particular Fenneropenaeus chinensis and Sagmariasus verreauxi IAGR. Thus, we did not reconstruct this figure. Thank you!   

Page 5, Figure 3: Homo sapines should be Homo sapiens.

Response: sorry for our mistake, revision has been made. Please see the revised figure 3.

Page 5, figure 4: Define column M in the legend.

Response: M is the DNA marker, we have added this information in the legend for figure 4. Please see line 143.

Page 6, section 2.5. It is not explained what Sp-vasa and Sp-Dsx are I would suggest to move lines 251-263 from the discussion to the beginning of section 2.5 so that the purpose and meaning of the experiment is clearer.

Response: changes have been made accordingly. Please see line 167-179.

Page 9, line 289: “…we have also identified another insulin-like receptor that is highly expressed in stage 3 testis,…”: this should be referenced if published or in press, or otherwise mentioned as submitted or unpublished results.

Response: According to your suggestion, we have rewritten this sentence to “In this regard, we have also identified another IR that is highly expressed in stage III tes-tis (unpublished results), suggesting that the regulation of spermatogenesis by insulin-like signaling in S. paramamosain is likely to be more complex.” Please see line 287-288.
